

# Chromosomal aberrations and prognostic analysis of secondary acute myeloid leukemia—a retrospective study

Mingzhu Song[1,2], Tun Zhang[3], Dongdong Yang[3], Hao Xiao[3], Huiping Wang[3], Qianling Ye[4] and Zhimin Zhai[3]

[1] Transfusion, The Affiliated Hospital of Anhui Medical University (Lu'an People's Hospital), lu'an, Anhui Province, China

[2] Anhui Medical University, Anhui Medical University, Hefei, Anhui Province, China

[3] Hematology, The Second Affiliated Hospital of Anhui Medical University, Hefei, Anhui, Hefei, Anhui Province, China

[4] Oncology, East Hospital Affiliated to Tongji University, Tongji University School of Medicine, Shanghai, Shanghai, China

Corresponding authors
Qianling Ye, yeqianling1@163.com
Zhimin Zhai, zzzm@889.com

## ABSTRACT

**Background**. Secondary acute myeloid leukemia (S-AML) patients generally have a poor prognosis, but the chromosomal aberrations of S-AML have been rarely reported. We aimed to explore the chromosomal aberrations and clinical significance in patients with S-AML.

**Patients and methods**. The clinical characteristics and karyotypes of 26 patients with S-AML were retrospectively analyzed. The overall survival (OS) was measured from the time of the patients' transition to AML (*i.e.*, at S-AML diagnosis).

**Results**. The study included 26 S-AML patients (13 males and 13 females), with a median age of 63 years (range, 20–77 years). They transformed from various hematologic malignancies or solid tumors; most of them were secondary to myelodysplastic syndrome (MDS). About 62% of the S-AML patients showed chromosomal aberrations. The serum lactate dehydrogenase (LDH) level in S-AML patients with abnormal karyotype was higher than those with normal karyotype. Apart from the differences in treatment regimens, S-AML patients with chromosomal aberrations had shorter OS ($P < 0.05$).

**Conclusion**. S-AML patients with abnormal karyotype have higher LDH levels and shorter OS than normal karyotype patients, and the OS of hypodiploidy was much shorter than hyperdiploid.

## INTRODUCTION

Secondary acute myeloid leukemia (S-AML) refers to AML developing either after a prior hematologic disorder, usually myelodysplastic syndrome (MDS), myeloproliferative neoplasms (MPN), or MDS/MPN (*Ossenkoppele & Montesinos, 2019*; *Leone et al., 1999*). Compared with newly diagnosed primary AML (P-AML), S-AML has a poorer prognosis, lower remission rates, and shorter OS (*Hulegårdh et al., 2015*; *Granfeldt et al., 2015*). S-AML

is usually common in elderly patients, which may be related to the high incidence of MDS and other malignant tumors in the elderly population. Although intensive chemotherapy regimens are adopted, S-AML patients prognosis is still poor, especially in elderly patients (*Szotkowski et al., 2010*).

Some predictors of transformation have been identified, including mutations in genes involved in growth-signaling pathways (NRAS, FLT3, etc.) and mutations commonly seen in AML (NPM1, WT1, IDH2) (*Bejar, 2018*). Recent advances in cytogenetic analysis have revealed that many chromosomal aberrations are associated with the onset and recurrence of AML (*Yamaguchi, 2020*). The recognition and understanding of chromosomal aberrations for the diagnosis and treatment of AML patients is of great significance (*Liu et al., 2020*). Chromosomal aberrations are likely to be associated with disease progression in S-AML (*Ayres-Silva et al., 2018*).

Some major clinical features are poor prognostic factors for AML. For instance, high WBC and/or LDH levels were identified as significant predictive features for OS (*Wan et al., 2019*; *Zhou et al., 2019*). Here, we analyzed the clinical and cytogenetic characteristics of 26 S-AML participants to explore the possible pathogenesis of S-AML patients further.

## MATERIALS & METHODS

### Participants

A total of 26 S-AML participants diagnosed or treated in the Second Affiliated Hospital of Anhui Medical University from January 2009 to January 2020 were collected. All the newly diagnosed S-AML participants met the 2008 or 2016 WHO criteria (*Vardiman et al., 2009*; *Arber et al., 2016*). In the WHO scheme, a myeloid neoplasm with 20% or more blasts in the PB or BM is considered to be AML, and in some cases associated with specific genetic abnormalities, the diagnosis of AML may be made regardless of the blast count in the PB or BM. Clinical characteristics of all the participants were obtained from medical records. The study was performed in accordance with the principles expressed in the Declaration of Helsinki. The Institutional Review Board of the Second Affiliated Hospital of Anhui Medical University approved this study, and the approval number was PJ-YX2019-008 (F2).

### Karyotype analysis

Of the 26 S-AML participants, 25 had a cytogenetic analysis performed at the time of progression to AML (*i.e.,* at S-AML diagnosis). All cytogenetic analyses were carried out in a standardized fashion at the Chromosome Laboratory, Department of Hematology, The Second Affiliated Hospital of Anhui Medical University. Bone marrow specimens were prepared by the short-term culture method and the G-banding method. Twenty (20) metaphase spreads were examined per patient, if available. The International System for Human Cytogenetic Nomenclature (ISCN) was used for karyotyping (*Simons, Shaffer & Hastings, 2013*). The S-AML participants were then divided into two groups: normal karyotype (NK) (chromosome number and structure were normal) and abnormal karyotype (number or structure abnormalities). According to the number of chromosomes,

the abnormal karyotype group was further subdivided into diploid (46 chromosomes), subdiploid (<46 chromosomes), and hyperdiploid (>46 chromosomes).

## Laboratory examination

The differences of some laboratory examination including haematological and biochemical parameters between the normal karyotype and abnormal karyotype were compared. Laboratory examination were obtained from medical records, including red blood cell (RBC) counts, white blood cell (WBC) counts, platelet counts (PLT), lymphocyte counts (LYM), mononuclear cell counts (MO), neutrophil counts (NEUT), hemoglobin (Hb), hypersensitive c-reactive protein (Hs-CRP) and lactate dehydrogenase (LDH), using the fully automated hematology analyzer Sysmex XE-2100 (Sysmex Corporation, Kobe, Japan) and the fully automated biochemical analyzer AU5831 (Beckman Coulter, Brea, CA, USA).

## Follow up

Participants were followed till death, loss to follow-up, or the end of the study follow-up period on July 20, 2020. OS was calculated from the time of S-AML diagnosis to the date of death or last follow-up. Medical record retrieval and telephone follow-up were performed during the study period.

## Statistical analysis

The student's $t$-test was used to test the differences between the two groups for quantitative and normally distributed variables; the Mann–Whitney U test was used for non-parametric variables. Kaplan–Meier survival curves were used to estimate OS. Statistical analyses were performed with the IBM SPSS 25.0 software. Results were considered significant at $p < 0.05$.

# RESULTS

## Participants characteristics

Half (n =13/26) of these 26 S-AML participants were men. In addition to unclassified AML (38.5%, n =10/26), M2 was the most common FAB subtype in 42.3% (n =11/26) of the participants, followed by M4 (7.7%, n =2/26), M3 (3.8%, n =1/26), M5 (3.8%, n =1/26), M7 (3.8%, n =1/26). As for the diagnosis prior to AML, 57.7% (n =15/26) of the participants were secondary to MDS (one of them was secondary to MDS, but coexisted with chronic lymphocytic anemia (CLL)), 3.8% (n =1/26) of the participants were secondary to myelodysplastic-myeloproliferative neoplasms (MDS/MPN), 11.5% (n =3/36) of the participants were secondary to chronic myeloid leukemia (CML), and 7.7% (n =2/26) of the participants were secondary to chronic myeloid monocytic leukemia (CMML), primary myelofibrosis (PMF) and rectal cancer respectively. 3.8% (n =1/26) of the participants were secondary to gastric diffuse large B cell lymphoma. The basic characteristics of 26 S-AML participants was shown in Table 1. Other clinical features were also collected, such as treatment, which is an important determinant of OS, as well as factors that are closely related to patient prognosis.

The treatment of AML is complex, and multiple subentities of AML often require different therapies. In our study, many participants were treated with decitabine in

**Table 1  The basic characteristics of 26 S-AML participants.**

| Characteristics | Participants (N = 26) |
|---|---|
| **Median age (range)** | 63 (20-77) |
| **Gender** | |
| Male | 13 (50.0%) |
| Female | 13 (50.0%) |
| **FAB subtype** | |
| M2 | 11 (42.3%) |
| M3 | 1 (3.8%) |
| M4 | 2 (7.7%) |
| M5 | 1 (3.8%) |
| M7 | 1 (3.8%) |
| Unclassified | 10 (38.5%) |
| **Diagnosis prior to AML** | |
| MDS | 15 (57.7%) |
| MDS/MPN | 1 (3.8%) |
| CML | 3 (11.5%) |
| CMML | 2 (7.7%) |
| PMF | 2 (7.7%) |
| Rectal cancer | 2 (7.7%) |
| Gastric diffuse large B cell lymphom | 1 (3.8%) |

**Notes.**

AML, acute myeloid leukemia; S-AML, secondary acute myeloid leukemia; FAB, French-American-British classification of leukemia.

combination with other regimens. Other optional regimens such as azacytidine, IA/IAG regimen, and intrathecal injection have also been used to treat participants, depending on the patient's condition. The detailed therapeutic regimen of 26 S-AML participants is shown in Table 2.

## Karyotype test results

In this study, a total of 16 participants had chromosome abnormalities, 50% (n =8/16) of the participants had chromosome 5 abnormalities, 43.8% (n =7/16) of the participants had chromosome 7-8 and 11-13 abnormalities, 37.5% (n =6/16) of the participants had chromosome 9 and 22 abnormalities. Chromosome 15 and 21 abnormalities were found in 31.3% (n =5/16) of the participants. The frequencies of the other types of chromosomes were relatively low. The detail of each chromosome abnormality was shown in Table 3. Chromosomal aberrations showed numerical and structural abnormalities in most chromosomes. Hypodiploidy and hyperdiploidy were the two most common genetic abnormalities in AML in this study, accounting for 11 cases (11/16) of abnormal chromosomes. The specific karyotypes of the 26 participants with clonal aberrations were listed in Table 4.

Song et al. (2023), *PeerJ*, DOI 10.7717/peerj.15533

**Table 2  The detailed therapeutic regimen of 26 S-AML participants.**

| No | Gender | Age | Original diagnosis | AML | Treatment (after the time of S-AML diagnosis) | Outcome (until July 20, 2021) | OS (Days) |
|---|---|---|---|---|---|---|---|
| 1 | Male | 72 | MDS | M7 | Decitabine alone | Death | 80 |
| 2 | Female | 56 | MDS | M2 | Decitabine+CAG(Ara-C, Aclarubicin, and G-CSF ), HAAG(Homoharringtonine, Ara-C, Aclarubicin, and G-CSF) | Death | 211 |
| 3 | Male | 76 | MDS | M2 | Decitabine+CAG(Ara-C, Aclarubicin, and G-CSF )+ATO | Death | 575 |
| 4 | Female | 65 | MDS | M4 | No treatment data available | Survival | 600 |
| 5 | Female | 66 | MDS | AML (un-classified) | No treatment data available | Death | 485 |
| 6 | Female | 62 | MDS | AML (un-classified) | CAG(low dose Cytarabine, Aclarubicin, and G-CSF )+ATO+EPO | Death | 55 |
| | | | | | IAG(idarubicin+Ara-C+G-CSF), DA(Daunorubicin+Ara-C) | | |
| | | | | | Azacitidine+HAG (Homoharringtonine, Ara-C, and G-CSF) | | |
| 7 | Female | 65 | MDS | M2 | intrathecal injection(MTX, DXM, and Ara-C) | Death | 108 |
| | | | | | Decitabine, thalidomide, ubenimex, Lenalidomide, Tretinoin, TPO | | |
| 8 | Male | 61 | MDS | AML (un-classified) | Decitabine+CAG(Ara-C, Aclarubicin, and G-CSF ) | Loss to follow-up | 10 |

Peerj

**Table 2** (*continued*)

| No | Gender | Age | Original diagnosis | AML | Treatment (after the time of S-AML diagnosis) | Outcome (until July 20, 2021) | OS (Days) |
|----|--------|-----|--------------------|-----|-----------------------------------------------|-------------------------------|-----------|
| 9 | Male | 70 | MDS | AML (un-classified) | Low dose Decitabine+EAG(epirubicin, Ara-C, and G-CSF) | Death | 105 |
| | | | | | Decitabine+MAG(mitoxantrone, Ara-C, and G-CSF) | | |
| | | | | | Decitabine+CMG(Ara-C, mitoxantrone, and G-CSF) | | |
| | | | | | Thalidomide | | |
| 10 | Male | 61 | MDS | AML (un-classified) | Decitabine+HAG (homoharringtonine, Ara-C, and G-CSF) | Survival | 210 |
| | | | | | Ubenimex, Tretinoin, azacitidine | | |
| 11 | Female | 77 | MDS | M2 | Tretinoin+ATO+decitabine+HAG (homoharringtonine, Ara-C, and G-CSF)+EAG(epirubicin, Ara-C, and G-CSF)+MAG(mitoxantrone, Ara-C, and G-CSF) | Loss to follow-up | 213 |
| 12 | Female | 20 | MDS | M2 | IA(Idarubicin, Ara-C) | Loss to follow-up | 150 |
| | | | | | Decitabine+CAG(Ara-C, Aclarubicin, and G-CSF )+ATO | | |
| | | | | | Decitabine+CHG(Ara-C, Homohar-ringtonine, and G-CSF)+ATO | | |
| 13 | Female | 66 | MDS | AML (un-classified) | No treatment data available | loss to follow-up | 60 |
| 14 | Male | 69 | MDS/MPN | M2 | Low dose Ara-C, in-terferon, and dasa-tinib | loss to follow-up | 60 |
| 15 | Female | 64 | MDS | M2 | CAG(Ara-C, Aclaru-bicin, and G-CSF )+decitabine | death | 226 |

**Table 2** (*continued*)

| No | Gender | Age | Original diagnosis | AML | Treatment (after the time of S-AML diagnosis) | Outcome (until July 20, 2021) | OS (Days) |
|---|---|---|---|---|---|---|---|
| 16 | Female | 30 | gastric diffuse large B cell lymphom | M3 | Tretinoin+ATO+intrathecal injection(MTX, DXM, and Ara-C) | survival | 1305 |
| 17 | Male | 46 | CML | AML (unclassified) | DA(Daunorubicin+Ara-C)Idarubicin | Death | 180 |
| | | | | | HAG (Homoharringtonine, Ara-C, and G-CSF) | | |
| | | | | | Dasatinib+Imatinib(Oral administration of dasatinib and imatinib was subsequently discontinued because of the T325I mutation, which suggested resistance to all tyrosine kinases), Hydroxycarbamide, etoposide, and ATO. | | |
| 18 | Male | 61 | CMML | M2 | IA(Idarubicin, Ara-C) | Death | 323 |
| | | | | | Decitabine+CAG(Ara-C, Aclarubicin, and G-CSF) | | |
| | | | | | Decitabine+HAG (homoharringtonine, Ara-C, and G-CSF)+Tretinoin+ATO | | |
| | | | | | Stanozolol, etoposide, ubenimex, and thalidomide | | |
| | | | | | Dorubicin liposomes and hexadecadrol | | |
| | | | | | Low dose methotrexate, and azacitidine | | |

Song et al. (2023), *PeerJ*, DOI 10.7717/peerj.15333

**Table 2** (*continued*)

| No | Gender | Age | Original diagnosis | AML | Treatment (after the time of S-AML diagnosis) | Outcome (until July 20, 2021) | OS (Days) |
|---|---|---|---|---|---|---|---|
| 19 | Female | 55 | PMF | M2 | Decitabine+IA(Idarubicin, Ara-C) Hematopoietic stem cell microtransplantation DAE (Doxorubicin+Ara-C+Etoposide) | Death | 143 |
| 20 | Male | 61 | MDS (co-exist with CLL) | AML (un-classified) | ATO+VP-16+Ara-C+G-CSF | death | 62 |
| 21 | Female | 66 | CMML | M4 | Decitabine+HAG (homoharringtonine, Ara-C, and G-CSF) Low dose Decitabine+ATO+DAG (Daunorubicin+Ara-C+G-CSF) Etoposide, Ara-C, and azacitidine | Loss to follow-up | 328 |
| 22 | Male | 38 | PMF | M5 | ME (Mitoxantrone, Etoposide), homoharringtonine, Ara-C, ATO | loss to follow-up | 450 |
| 23 | Male | 72 | rectal cancer | M2 | Decitabine+CAG(Ara-C, Aclarubicin, and G-CSF) CTK cell infusion G-CSF, Ara-C, ATO Decitabine+darubicin or Pirarubicin +Ara-C | survival | 2560 |
| 24 | Female | 67 | Rectal cancer | AML (un-classified) | Decitabine+Ara-C | Death | 21 |

Song et al. (2023), *PeerJ*, DOI 10.7717/peerj.15333

**Table 2** (*continued*)

| No | Gender | Age | Original diagnosis | AML | Treatment (after the time of S-AML diagnosis) | Outcome (until July 20, 2021) | OS (Days) |
|---|---|---|---|---|---|---|---|
| 25 | Male | 32 | CML | AML (un-classified) | MA(Mitoxantrone, and Ara-C) | death | 270 |
| | | | | | CAG(Ara-C, Aclarubicin, and G-CSF) | | |
| | | | | | Dasatinib, methotrexate | | |
| | | | | | Intrathecal injection(MTX, DXM, and Ara-C) | | |
| 26 | Male | 44 | CML | M2 | No treatment data available | loss to follow-up | 5 |

**Table 3  The detail of each chromosome abnormality in 26 S-AML participants.**

| Chromosome | | Participants ($N = 26$) |
|---|---|---|
| | Normal chromosome | 62.5% (n=10/16) |
| | 5 | 50.0% (n=8/16) |
| | 7 | 43.8% (n=7/16) |
| | 8 | 43.8% (n=7/16) |
| | 11 | 43.8% (n=7/16) |
| | 12 | 43.8% (n=7/16) |
| | 13 | 43.8% (n=7/16) |
| | 9 | 37.5% (n=6/16) |
| | 22 | 37.5% (n=6/16) |
| | 15 | 31.3% (n=5/16) |
| | 21 | 31.3% (n=5/16) |
| | **2** | 25.0% (n=4/16) |
| | **3** | 25.0% (n=4/16) |
| | 17 | 25.0% (n=4/16) |
| Abnormal chromosome | 19 | 25.0% (n=4/16) |
| | 20 | 25.0% (n=4/16) |
| | 34 | 18.8% (n=3/16) |
| | 14 | 12.5% (n=2/16) |
| | 18 | 12.5% (n=2/16) |
| | 23 | 12.5% (n=2/16) |
| | 25 | 12.5% (n=2/16) |
| | 6 | 6.3% (n=1/16) |
| | 26 | 6.3% (n=1/16) |
| | 29 | 6.3% (n=1/16) |
| | 31 | 6.3% (n=1/16) |
| | 32 | 6.3% (n=1/16) |
| | 36 | 6.3% (n=1/16) |
| | 37 | 6.3% (n=1/16) |
| | X | 6.3% (n=1/16) |
| | Y | 6.3% (n=1/16) |

## Karyotypes and laboratory investigation

The results showed that LDH level was statistically higher in participants with S-AML with chromosomal aberrations ($P < 0.05$). The scatter diagram for the LDH levels between the 2 groups is shown, which only contains 23 participants (Fig. 1). RBC, WBC, PLT, and other laboratory examination results showed no significant difference between the normal and abnormal karyotype groups (Table 5).

## Overall survival (OS)

Except for participants who were lost to follow-up, there were two deaths in the normal karyotype group and 12 deaths in the abnormal group. The median OS of normal and abnormal karyotypes was 212 days and 162 days, respectively. In addition, the abnormal karyotype included seven hyperdiploid and five hypodiploid participants, and the median

**Table 4  Chromosome karyotypes of the 26 S-AML participants.**

| Karyotypes (N) | | Chromosome of S-AML |
|---|---|---|
| Normal (10) | Diploid (10) | 46,XY |
| Abnormal (16) | Diploid[#] (7) | 46,XY,-7,+marker.[10] |
| | | 46,XX[3]/46,XX,+der(8)del(q22),del(12)(p11),-2,-5,-7,-11,-17,+22,+marker*3[17] |
| | | 46,XY,del(5)q(23),add(17)p(12),-9,12,20,marker ×3.[5] |
| | | 46,XY,t(9;22)(q34;q11),t(2;12;15),(p13;q13;p11),+8.[20] |
| | | [a]46,XY,t(9;22)(q34;q11)[8]/46,XY,t(9;22)(q34;q11),ins(3;3)(q25;q21q25)[5] |
| | | 45,XY,add(3)(q29),del(5)(q23),add(12)(p15),-7.[20] |
| | | [#] 43-46,XX.-2,-3,?add(3)(q11),del(5)(q13q31),del(7)(q31),add(11)(p15),-15,-17,add(17)(p13),-18,add(19)(p13),add(22)(q13)+mar,inc[cp20]. |
| | hypodiploid[*] (5) | 43,X,t(5;19)(q21;q13),7q+,-7,-12,-20,-Y,+marker.[7]/44,XY,5q-,7q+,-12,-18,-20,+marker1,+marker2.[13] |
| | | 45,XY,-5,-9,+mar[7]/45,XY,del(5)(q15),-9,add(11)(q25)[4]/44,XY,add(5)(p15),del(5)(q15),del(7)(q11),der(12)del(12)(p12)add(12)(p12),-13,-19,-21,+mar[5] |
| | | [*]40-48 XX,add(1)(p36),add(2)(q37),del(5)(q15),add(12)(p13),-8,-9,-11,-22,+marker ×3.inc.[cp15] |
| | hyperdiploid[*] (7) | 47,XX,+8.[20] |
| | | 48,XXX,del(20)(q13),+X,+marker.[8]/48,XX,del(20)(q13),+14,+marker.[3] |
| | | 48,XY,inv(3)(q21q26),+8,t(9;22)(q34;q11),i(17)(q11),+der(22)t(9;22)(q34;q11)[20] |
| | | 48,XY,20q-,+8,+13.[5] |
| | | 48,XX,t(1;?)(q21;?),+der(1)t(1;?)(p32;?),-6,-7,+14,+19,+r.[8]/48,XX,t(1;?)(q21;?),+der(1)t(1;?)(p32;?),-6,-7,+14,+19,+marker.[2] |
| | | 47,XY,5q-,+8.[15] |

**Notes.**
[a]the chromosome of the patient was collected at primary diagnosis (2 months ago);
[*]the chromosome contains in all the three kinds of abnormal karyotypes of chromosome;
[#]the chromosome contains in both diploid and hypodiploid of abnormal karyotypes of chromosome.

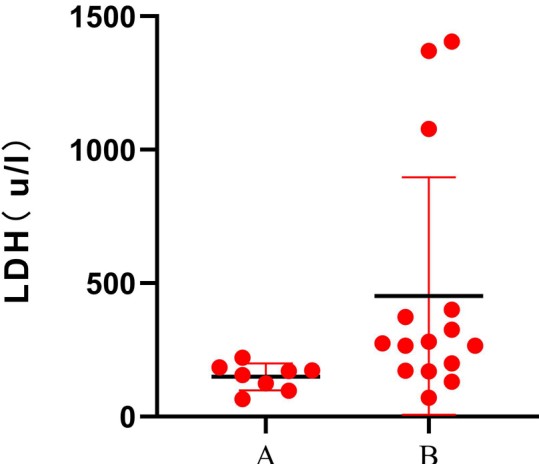

**A=Normal chromosome karyotypes(n=8)**
**B=Abnormal chromosome karyotypes(n=15)**

**Figure 1** LDH level in normal and abnormal chromosome karyotypes.

**Table 5** Laboratory examination in normal and abnormal chromosome karyotypes.

| Laboratory examination | Normal chromosome karyotypes ($n = 10$) median (range) | Abnormal chromosome karyotypes ($n = 16$) median (range) | P |
|---|---|---|---|
| RBC ($\times 10^{12}$/L) | 2.16(1.47–3.94) | 1.98(1.38–5.49) | 0.551 |
| WBC($\times 10^{9}$/L) | 1.96(0.3–11.13) | 3.28(0.33–47.17) | 0.391 |
| PLT($\times 10^{9}$/L) | 13.5(3–269) | 28.5(5–207) | 0.262 |
| LYM($\times 10^{9}$/L) | 1.09(0.27–2.82) | 0.82(0.14–22.08) | 0.816 |
| MO($\times 10^{9}$/L) | 0.14(0–2.09) | 0.41(0–15.89) | 0.165 |
| NEUT($\times 10^{9}$/L) | 0.34(0–9.35) | 1.72(0.02–36.62) | 0.182 |
| Hb(g/L) | 66.5(44–121) | 64(49–152) | 0.363 |
| hsCRP(mg/L) | 61(0.3–87.2)[a] | 39(1.5–193.7) | 0.452 |
| LDH(U/L) | 163.5(65–220)[b] | 274(71–1406)[c] | 0.008 |

Notes.
[a] $n = 9$
[b] $n = 8$
[c] $n = 15$

OS was 211 days and 62 days, respectively. The Kaplan–Meier (KM) survival curve results showed that the OS of S-AML participants with abnormal karyotypes was shorter than those with normal karyotypes ($P = 0.038$) (Fig. 2). Also, compared with normal karyotypes, the OS of hyperdiploid was shorter, while the OS of hypodiploidy was much shorter ($P = 0.038$) (Fig. 3). KM survival curves were plotted for specific chromosomes with more than five chromosomal abnormalities, as shown in Fig. 4, so as to measure the OS rate by observing different chromosomal aberrations separately. The results showed that the OS of chromosome 8 and 13 abnormalities was not statistically significant compared

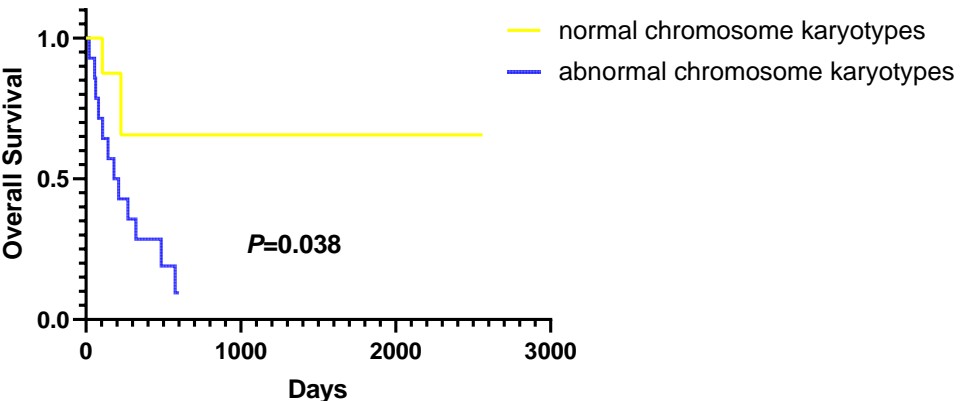

**Figure 2** OS in normal and abnormal chromosome karyotypes of S-AML participants.

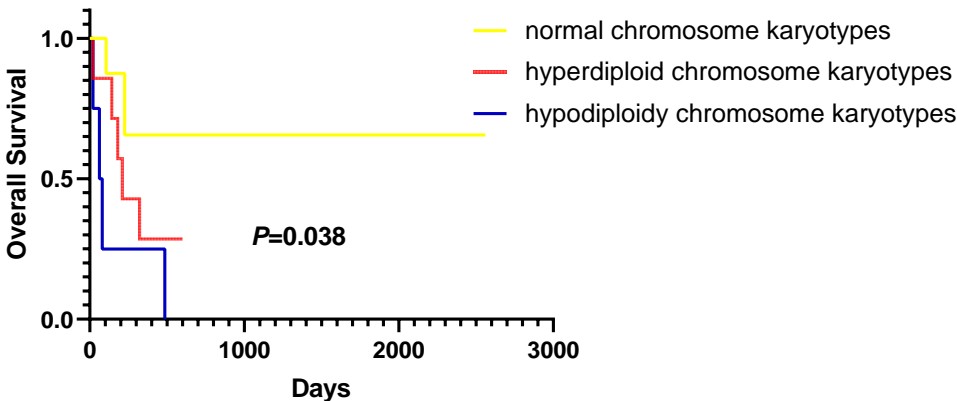

**Figure 3** OS in normal, hyperdiploid and hypodiploidy chromosome karyotypes of S-AML partici-pants.

with normal chromosomes ($P > 0.05$). However, the OS of chromosomes 5, 7, 9, 11, 12 and 22 was statistically different from that of the normal chromosome ($P < 0.05$).

## DISCUSSION

S-AML is a heterogeneous disease; its incidence increases with age, but therapy remains a challenge (*Collinge et al., 2018*). Myelodysplastic syndrome (MDS) is characterized by cytopenia, osteomyelodysplasia, hematopoietic dysfunction, and a high risk of transition to AML (*Menssen & Walter, 2020*). More than half of the S-AML participants reported in this study transformed from MDS to AML. Compared with primary AML patients (P-AML), S-AML patients have a worse clinical prognosis regarding complete remission rate (CR), recurrence-free survival rate, and OS rate (*Cheung et al., 2019*). Many factors can cause the poor curative effect, poor prognosis, and short survival time of S-AML patients. Our previous study showed that abnormally increased peripheral blood regulatory T cells

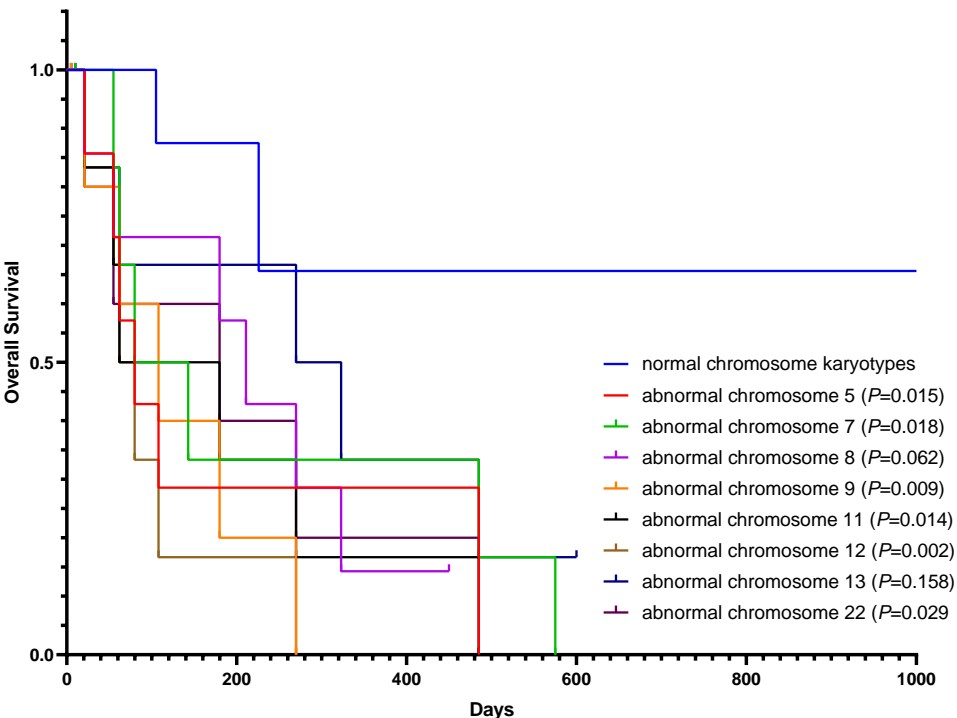

**Figure 4** OS in normal and different chromosome aberrations of S-AML participants.

(Treg) might cause an imbalance in the immune status of S-AML patients, which might be relevant to the poor chemotherapy response and short survival time of S-AML patients (*Ye et al., 2020*). There is growing evidence that chromosomal aberrations represent a common genomic imbalance of cancer and are associated with cancer prognosis and response to chemotherapy and immunotherapy (*Kou et al., 2020*). Common genomic abnormalities, such as deletions, polyploidy, translocations, and complex chromosomal aberrations, are common chromosomal alterations, which may be associated with altered microRNAs expression patterns, as many deregulated microRNAs localize to aberrant chromosomal regions (*Bauer et al., 2020*). It has been reported that there are tumor suppressor genes on chromosome 6q, 7p, 10p, 11q, 14q, and 20q, which is essential for the transformation from MDS to AML (*Mori et al., 2000*). Chromosomal aberrations are common in hematological malignancies. Larson (*Larson, 2007*) have shown that the characteristics of cytogenetic abnormalities in S-AML are similar to those in P-AML. However, compared with P-AML, S-AML patients' prognosis is worse; S-AML patients also have a higher frequency of adverse and moderate risk chromosomal aberrations.

Chromosomal aberrations are associated with progression to S-AML and deserved further study. The purpose of this study was to analyze the chromosomal aberrations of S-AML participants and further explore the factors connected with the survival and prognosis of S-AML in combination with relevant laboratory examinations. Our results indicated that most S-AML participants had abnormal karyotypes, including autosomal

and sex chromosome aberrations. Abnormal changes in autosomal karyotypes were more common in S-AML participants and were closely related to survival and prognosis. Studies have demonstrated an increased incidence of abnormalities on chromosomes 5 and 7 in patients with S-AML (*Mannan et al., 2020*; *Seymour et al., 1999*). In our study, 62.5% (n =10/16) abnormal karyotypes had aberrations on chromosomes 5 and 7. Admittedly, our sample size and the data were limited; we could not get much information based on the results of the 26 S-AML participants. Abnormal changes of sex chromosomes have been rarely reported in myeloid malignancies (*Gurnari et al., 2018*). We found an extra sex chromosome (X chromosome) in an elderly woman (65 years old) with FAB-M4 who transformed from MDS; the abnormal karyotypes were: 48,XXX,del(20)(q13),+X,+marker.[8]/48,XX,del(20)(q13),+14,+marker.[3]. The patient was alive at the end of the study follow-up period. Recently, a report associated the X chromosome loss with a better prognosis in female AML patients with t (8;21) (*Chen et al., 2020*). We also detected Y chromosome deletion in an elderly male (61 years old) patient who progressed from MDS; the abnormal karyotype was: 43,X,t(5;19)(q21;q13),7q+,-7,-12,-20,-Y,+marker.[7]/44,XY,5q-,7q+,-12,-18,-20,+marker1,+marker2.[13]. Unfortunately, the patient was lost to follow-up, and we do not know whether the patient is alive or not. Some studies have suggested that Y chromosome loss is an age-related phenomenon with no prognostic significance (*Zeidan & Phatak, 2008*). Another study also indicated that Y chromosome loss increases with age, but it reduces the risk of transformation from MDS to leukemia (*Nomdedeu et al., 2017*). In contrast, the loss of Y chromosome was associated with a high recurrence/relapse rate in AML male patients with t (8:21) (*Chen et al., 2020*). The relationship between sex chromosome aberrations and survival in S-AML patients needs to be further explored on larger cohorts.

Most AML patients with chromosome number abnormalities may manifest with an increase of 1–2 chromosomes (47–48 chromosomes), known as low hyperdiploid, or rare high hyperdiploidy (49–65 chromosomes), both of which are associated with poor outcome in AML (*Lazarevic et al., 2015*; *Luquet et al., 2008*; *Chilton et al., 2014*); (*Sandahl et al., 2014*). *Holmfeldt et al. (2013)* reported no difference in 5-year OS and EFS (event-free survival) between AML patients with non-hyperdiploid and hyperdiploid karyotypes (48-65 chromosomes). Hypodiploidy (<46 chromosomes) has been reported mostly in acute lymphoblastic leukemia (ALL) but rarely in AML (*Holmfeldt et al., 2013*; *Pui et al., 2019*; *Peterson et al., 2019*). However, there is a current lack of further research on the prognosis and survival in S-AML patients with hypodiploidy or hyperdiploidy. Studies have shown that the feature of hypodiploidy usually involve abnormalities on chromosomes 5, 7, or 17, with structural aberrations t(8;21)(q22;q22), and is usually accompanied by clonal loss of sex chromosomes (*Yeh & Tirado, 2021*). This is consistent with the characteristics of hypodiploidy in our study. All the subdiploid karyotypes in our study had a deletion or partial deletion of chromosome 5, 7, or 17, and one patient also had a concomitant Y chromosome loss. Studies have shown that the common chromosomal abnormalities in hyperdiploidy are chromosome 8, 21 and 19 (*Veigaard, Norgaard & Kjeldsen, 2011*). We identified the most abnormalities on chromosome 8 in hyperdiploidy in our study, and the

hyperdiploid group also had sex chromosome abnormalities. This leads us to hypothesize whether sex chromosome abnormalities are more likely to occur in hypodiploidy and hyperdiploidy. Given the small amount of data in this study, more chromosome data of S-AML are needed for further exploration.

In addition to other factors affecting OS, such as various treatment regimens, our research found that karyotypes were closely related to S-AML participants' survival; participants with abnormal karyotypes demonstrated inferior OS compared with those with normal karyotype. What is more, S-AML participants with hypodiploidy showed worse outcomes than those with hyperdiploidy.

LDH not only plays a vital role in the early diagnosis and prognosis of many solid tumors but also plays a crucial role in evaluating the severity of leukemia patients (*Koukourakis et al., 2005*; *Banescu et al., 2019*). LDH positively correlated with tumor burden and is an independent prognostic factor for early death in hyperleukocytic AML (*Piccirillo et al., 2009*). Our results showed a significantly increased LDH level in the abnormal karyotype group than the normal group. It suggests that the higher the LDH level in S-AML patients, the greater the tumor burden, the greater the possibility of karyotype abnormality, and the worse the OS rate. LDH is a valuable enzyme among many biochemical parameters and can be easily detected routinely in many clinical laboratories. In brief, abnormalities of LDH and karyotypes are closely related to the severity, survival, and prognosis of S-AML patients that can be a very valuable indicator for further risk stratification of S-AML in the future. Due to the small sample size of this study, the results of this study may have a certain bias, and the relationship between LDH and chromosome karyotype needs to be further studied.

It is of great significance to choose the appropriate chemotherapy regimen to manage patients with AML effectively. In clinical practice, an individualized treatment regimen is often tailored to the patient's tolerance and other specific conditions. In our study, many participants were treated with decitabine in combination with other regimens. Decitabine is a demethylation agent that is effective and safe in older patients with AML; its combination with other regimens (*e.g.*, CAG (Ara-C, Aclarubicin, and G-CSF), retinoic acid) results in a higher OS rate than decitabine alone (*Bian et al., 2019*). It is too unspecific for the treatment regimens related specifically to the 26 S-AML participants in our study, which may be an important factor affecting OS.

There are some limitations to our study. Firstly, the abnormality of sex chromosomes may relate to the survival and prognosis of S-AML, but no definite conclusion could be drawn because of the small number of sex chromosome aberrations in our study. A small number of patients did not get 20 metaphase spread, and there may be chromosomal abnormalities that have not been found, leading to a certain bias in the results. Apart from this, the accurate information of all participants could not be obtained through telephone follow-up in this study, which may interfere with the experimental results. Additionally, this study is a single-center retrospective study; the number of included cases was relatively small, so further study expanding the sample size is needed to validate our results. Moreover, with the heterogeneity of the individualized treatment among AML

participants, the treatment regimens could constitute an important source of limitation, which may have influenced the results.

## CONCLUSIONS

In conclusion, our research highlights chromosomes and LDH contributions to the poor prognosis of S-AML participants. Also, the abnormality of sex chromosomes may be associated with the survival and prognosis of S-AML participants. Understanding the multifactorial contributions will lead to more precise risk classification and treatment strategies. More factors related to the survival and prognosis of S-AML need to be explored, which may contribute to monitoring the progression of the disease, early diagnosis, and improved treatment.

### Funding
This article was funded by the National Natural Science Foundation of China, grant number 81602914. The funders had no role in study design, data collection and analysis, decision to publish, or preparation of the manuscript.

### Grant Disclosures
The following grant information was disclosed by the authors:
National Natural Science Foundation of China: 81602914.

### Competing Interests
The authors declare there are no competing interests.

### Author Contributions
- Mingzhu Song analyzed the data, prepared figures and/or tables, authored or reviewed drafts of the article, and approved the final draft.
- Tun Zhang analyzed the data, prepared figures and/or tables, and approved the final draft.
- Dongdong Yang performed the experiments, prepared figures and/or tables, and approved the final draft.
- Hao Xiao analyzed the data, prepared figures and/or tables, and approved the final draft.
- Huiping Wang analyzed the data, authored or reviewed drafts of the article, and approved the final draft.
- Qianling Ye conceived and designed the experiments, authored or reviewed drafts of the article, and approved the final draft.
- Zhimin Zhai conceived and designed the experiments, authored or reviewed drafts of the article, and approved the final draft.

### Data Availability
The raw measurements are available in Tables 1–4.

## Supplemental Information

Supplemental information for this article can be found online at http://dx.doi.org/10.7717/peerj.15333#supplemental-information.

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
