# Peer review of "Chromosomal aberrations and prognostic analysis of secondary acute myeloid leukemia—a retrospective study"

_PeerJ, doi:10.7717/peerj.15333_

## Round 0.1 · original submission · Minor Revisions

The report submitted by the authors is generally well-written and constructed. However, they should choose between a) amending the title to accomodate the fact that they worked with a very small cohort of patients (26); or b) going through further analysis as suggested by one of the reviewers.

The authors should be commended for the analysis of the data that seem correctly done and comprehensive; providing the therapeutic regimen of the 26 S-AML patients is also important, particularly for potential translations into the clinic. However, karyotypes (table 3) should be further explained/analyzed or at least discussed.

The manuscript should go through minor-to-medium revisions before acceptance for publication. Answers to the reviewers' comments should also be provided. Please, inform the editorial in case extra time is needed for these revisions. Please, also make sure that all the acronyms are properly address in the text/legends and all references are properly indicated or matched to the citation. Make sure to proofread before final submission.

·

Basic reporting

• Although the manuscript is written in professional English language, there are sections that needs to be reviewed to improve sentence construction. Please see general comments for examples.
• The introduction is short and concise with proper references, however it lacks other reasons for transformation or development of S-AML. Please see the general comments section for suggestions to improve this.
• Structure conforms to journal specifications.
• Figures and tables are of fair quality. Please see the general comments section for specific issues that needs to be addressed.
• Raw data was supplied and sufficient.

Experimental design

• The study conforms to the scope of the journal and contributes to the field of AML in general. Although I would propose that the authors promotes this study as a pilot study due to the low participant numbers. Also, authors need to provide better justification to why there is such a low number of S-AML cases over such a long period.
• Research rationale and aim was clearly stated and links to the literature gaps associated with S-AML.
• Investigations performed conformed to International guidelines and ethical standards. A major gap/limitation is the follow-up that was done telephonically. How was this conducted, was a specific questionnaire used, what measures were taken to eliminate bias from participant to participant answers, for instance was the experimental design of such to test participants responses (Same questions asked in different ways for example).
• Methods allow for replication of the study. Please refrain to repeat methodology in the results section. Further comments are listed in the general comments section of the review.

Validity of the findings

• Findings of the study was well presented and meaningful discussions were made with proper comparison to published studies. The authors also clearly indicates the limitations of the study and makes recommendations for future studies. I would recommend that further emphasize should be place on the LDH and chromosomal aberration results due to the small sample size. This again links to my previous comment that this study should be presented as a pilot study.
• The authors should be commended on the amount of data that is presented for the small cohort, which allows proper overview of the study population.
• Conclusions made were fair and limitations were again well described.

Additional comments

Line 29: ‘Rarely reported’ does not fit into the context. Rather refer to limited data or publications.

Page 31: Please indicate the number of participants after a % is used, example 62% (n=x/y). Please do this throughout the manuscript.

Line 54: Fix spacing issues.

Although chromosomal aberrations are correlated with S-AML, the introduction lacks reference to other genetic abnormalities that can lead to S-AML. It needs a short discussion in the literature. Furthermore, briefly discuss how treatment regimens can lead to S-AML.

Line 68: Please refer to the specific section in the WHO that addresses S-AML, merely stating WHO criteria is not sufficient.

Line 77: Chromosome Laboratory, Department of Hematology – Please indicate at which institution.

Line 79: ‘metaphase spreads were examined per patient, if available.’ What technique/method was used when 20 metaphase spreads were not available? This needs to be addressed properly as it could introduce a bias in the results.

Line 87-88: Please rephrase this sentence – some laboratory examinations is too unspecific. It seems that haematological and chemical parameters were investigated.

Preferably it is advisable to refer to participants of a study rather than patients.

Line 105: ‘with the IBM SPSS 25.0.’ Please add software after 25.0

Line 109: The line is repetitive, please remove.

Line 111-115: Please refer to table 1 in these lines.

Line 115: Space required between table and 1.

Line 118-124: This content belongs in the literature review and not in the results section. It is too unspecific for the treatment related specifically to the 26 S-AML participants in this study. Rather refer to table 2 and provide the prominent results from the table briefly.

Line 132-135: Please remove this section. It adds no value to the main text of the manuscript and can be viewed in table 3 in detail. Again, only highlight significant results from table 3 in the text.

Line 136: Change examination to investigation.

Line 137-138: Remove, this was mentioned in methods already.

Line 140: The scatter plot only contains LDH levels for 23 participants. Please disclose this in the text.

Line 144-148: Please have this section grammatically reviewed. The sentence construction needs attention.

Line 150-153: Figure 2 and Figure 3: Please double-check the p values between the two figures, or is it only coincidence that they are exactly the same?

Manuscript was well prepared, and context is provided in brackets to specific nomenclature which is excellent.

Please use articles (The and A) in the beginning of sentences.

Table 2: If there is no treatment regimen available, please consider using the following phrase: No treatment data available.

·

Basic reporting

The article is written in clear language, except some minor grammatical issues at places. They are highlighted in the attached annotated version. References are appropriately cited. Results are described in sufficient details as far as author's research objectives are concerned - to observe the chromosomal aberrations in S-AML and document this information. Authors did not go into any details regarding why these chromosomal changes exist or what could be the implications of these changes to S-AML.

Experimental design

This is mostly a descriptive article talking about a top-level overview of chromosomal alterations in S-AML. In summary, authors described ~62% of S-AML patients have some kind of chromosomal aberration. Next, they performed simple correlation analysis of these chromosomal aberrations with overall survival and some metabolic measurements.

They suggest that LDH levels correlate with chromosomal aberrations and patients with aberrations have lower overall survival. However, I feel the study has some limitations which would prevent implementation of author's observation in the field: 1) The sample size is not large enough. 2) The correlation between LDH and chromosomal aberrations does not seem particularly strong (see my additional comments). 3) Authors did not go into any details of their observations. For instance, authors found many different types of chromosomal aberrations, but in the end all of them were pooled together to make inference. Overall, it appears to me that the study could benefit from additional data analysis into the diversity of chromosomal aberrations observed and its connection to individual patient metrics. Additionally, authors could also validate their findings using samples from databases such as TCGA etc.

Validity of the findings

All the data is provided and statistics is well performed. Conclusions make sense to me.

Additional comments

1. Figure 1: I agree that the median LDH levels do look significantly different between group A and B. However, upon looking at the data points in group B, it seems only 3 points are driving the median higher. However, majority of points are pretty similar to group A. Given this observation, what was the rationale to conclude that group B had significantly higher LDH levels than group A. Most of the patients in group B might end up with LDH similar or within the error range of group A.

2. Would it be possible to measure overall survival by looking at different chromosome aberrations separately? For example, OS of patients with aberrations in chromosome 5 only etc.

3. Authors mention in the conclusions that chromosome aberrations + LDH contribute to poor prognosis of S-AML patients. It would be much more convincing if authors could confirm their conclusion by testing additional patient samples from databases. I am not sure about it, but authors could look into TCGA for similar or almost similar data and attempt to validate whether chromosome patterns and LDH levels correlate with OS.

4. Figure 2 and 3: It could be beneficial if authors could show the overall survival for each patient and group those line plots into color coded categories describing chromosomal aberration (such as normal, hyperdiploid, hypodiploid, chromosome number impacted etc.)

---

## Round 0.2 · accepted · Accept

The manuscript could however benefit from further language editing for which I suggest proofreading the final version by independent colleagues and/or professional speakers.
Congratulations!

·

Basic reporting

As per my previous review, the authors have amended my concerns relating to sentence construction.
Other comments were also addressed.

Experimental design

The experimental design was well explained and the study title was amended to include 'pilot study' as per my recommendations.
The low number of study participants were also addressed.

Validity of the findings

The prominent LDL findings were added to the discussion section as per my previous comments.

Additional comments

The authors addressed all the specific comments of concern that I had in the initial review of this manuscript.

·

Basic reporting

The authors have comprehensively responded to my queries. Authors have also added that this is a pilot study and larger study would be conducted in the future. Considering this, I have no further issues with this work.

Experimental design

The authors have comprehensively responded to my queries. Authors have also added that this is a pilot study and larger study would be conducted in the future. Considering this, I have no further issues with this work.

Validity of the findings

The authors have comprehensively responded to my queries. Authors have also added that this is a pilot study and larger study would be conducted in the future. Considering this, I have no further issues with this work.

Additional comments

The authors have comprehensively responded to my queries. Authors have also added that this is a pilot study and larger study would be conducted in the future. Considering this, I have no further issues with this work.